# Targeting NLRP10 in Atopic Dermatitis: An Emerging Strategy to Modulate Epidermal Cell Death and Barrier Function

**DOI:** 10.3390/ijms26199623

**Published:** 2025-10-02

**Authors:** Yi Zhou

**Affiliations:** Inflammation Research, Amgen Inc., South San Francisco, CA 94080, USA; yzhou04@amgen.com

**Keywords:** NLRP10, NLR, atopic dermatitis, cell death, epidermal barrier function, p63

## Abstract

Atopic dermatitis (AD) is the most common chronic inflammatory skin disease, characterized by pruritic and eczematous lesions. Skin barrier dysfunction and aberrant inflammatory responses are hallmark features of AD. Recent genome-wide association studies have implicated NLRP10, a unique member of the NOD-like receptors (NLRs) lacking a leucine-rich repeat (LRR) domain, in AD susceptibility. Unlike other NLRs, the physiological role of NLRP10 in skin remains incompletely understood. Emerging evidence shows that NLRP10 regulates keratinocyte survival and differentiation, acts as a molecular sensor for mitochondrial damage, enhances anti-microbial response and contributes to skin barrier function. This review summarizes current insights into NLRP10′s functions in skin homeostasis, its interplay with cell death pathways, and its role in maintaining skin barrier function. Furthermore, therapeutic opportunities to target NLRP10 as a novel strategy for modulating epidermal cell death and restoring barrier function in AD are highlighted.

## 1. Introduction

Atopic dermatitis (AD) is a very common chronic skin disorder that affects millions of people globally [1]. AD is characterized by intense itching, recurrent eczematous lesions and skin inflammation, significantly impacting quality of life and contributing to economic burden [2]. It is a heterogeneous medical condition and results from complex interactions of environmental and genetic factors [3,4,5]. Skin barrier dysfunction is regarded as a primary event and a major factor for AD pathogenesis [3,6].

The stratum corneum, the outermost layer of the epidermis, is fundamental for establishing skin barrier function. The stratum corneum is established through the terminal differentiation of keratinocytes into corneocytes, a process known as cornification. This leads to the formation of a protein- and lipid-rich layer that forms the permeability barrier and prevents passive water loss [7]. The epidermal barrier function also protects the skin by restraining the penetration of environmental allergens and microbial pathogens [6]. Skin barrier dysfunction may result from several pathogenic mechanisms, including immune dysregulation; impairments in terminal keratinocyte differentiation, such as filaggrin (*FLG*) deficiency caused by genetic mutation; reduced expression of antimicrobial peptides (AMPs); alterations in the composition of stratum corneum intercellular lipids; and dysbiosis of the cutaneous microbiome [8].

Recent large-scale multi-ancestry genome-wide association meta-analysis has identified more than 90 genetic loci associated with AD, including variants near *FLG* gene and many others [9,10]. Enrichment analysis of GWAS signals highlights pathways involved in immune regulation. Notably, genes in the interleukin-JAK-STAT pathway have been identified as causal candidates [9,10,11,12]. This aligns with clinical success of biologics blocking IL-4 and IL-13 and small molecule JAK inhibitors [13], validating these pathways as tractable drug targets in AD. Although causality of other implicated genes and their precise mechanisms require further investigation, these genetic studies offer potential drug candidates to bridge the gap between disease biology and therapeutic innovation in AD.

Variants at the NLRP10 locus, a gene in the NLR family proteins involved in innate immune response and epidermal barrier integrity, have also been linked to AD [9,14,15,16,17]. Emerging evidence implicates context-dependent human NLRP10 function in regulating inflammasome activation, influencing keratinocyte cell death, promoting anti-bacterial response and supporting epidermal barrier functions. This review summarizes recent advances in our understanding of physiological functions of human NLRP10 and its relevance to AD. These recent discoveries underscore the potential of NLRP10 as a promising therapeutic target for regulating keratinocyte cell death and restoring skin barrier function in AD.

## 2. NLRP10 and Other NLRs in Skin Homeostasis and Disease

At the forefront of host defense and inflammation in the skin are the NOD-like receptors (NLRs), a class of intracellular pattern recognition receptors (PRRs) [18]. At least 22 human NLRs have been identified based on the molecular similarity to disease resistant genes in plants. These were initially referred to as the CATERPILLER (CARD, transcription enhancer, R-binding, pyrin, lots of leucine repeats) gene family [19,20]. NLRs usually contain three core modules that contribute to the specificity and activity: (1) an N-terminal variable effector domain critical for protein–protein interaction and downstream signaling; (2) a central NACHT domain with ATPase activity for protein oligomerization; and (3) a C-terminal LRR domain for ligand sensing and autoregulation [21]. The unique tripartite architecture is critical for NLRs to recognize a wide repertoire of pathogen-associated molecular patterns (PAMPs) and damage-associated molecular patterns (DAMPs) to trigger innate immune activation. One exception to this general rule is NLRP10 (also known as PYNOD, NALP10, CLR11.1, NOD8, and PAN5), the only NLR member without the C-terminal LRR domain typically required for ligand sensing, suggesting an alternative role in modulating immune signaling (Figure 1A).

As critical components of the innate immune system, NLRs have diverse expression pattern across different tissues, of which many exhibit an enriched expression in circulating immune cells in blood [21]. A comprehensive analysis of the expression profiles of all 22 human NLRs across various tissues with the Genotype-Tissue Expression (GTEx) database [22] reveals that NLRP10 expression is restricted to the skin (Figure 1B). NLRP10 protein has also been confirmed to be detectable in epidermal keratinocytes and dermal fibroblasts, but not in peripheral blood mononuclear cells (PBMCs) [4,23,24,25]. NLRP10 expression is upregulated during keratinocyte differentiation and cornification, and it is highly expressed in terminally differentiated keratinocytes within the stratum granulosum [4,24,25]. The specific expression of NLRP10 in skin highlights its important roles in regulating skin homeostasis.

An extensive analysis of the roles of various NLRs in host defense and innate immunity has been thoroughly reviewed elsewhere [18,21,26]. Table 1 summarizes roles of major NLRs that are expressed in human skin and their relevance to skin diseases.

CIITA and NLRC5 are transcriptional co-activators for MHC class II and class I genes, respectively [26,27,28], thereby facilitating adaptive immune surveillance in the skin. NLRX1, predominantly localized to mitochondria, modulates reactive oxygen species and apoptotic pathways while dampening antiviral signaling through the RIG-I/MAVS axis, restraining excessive inflammation [29]. Several NLRs act as cytosolic PRRs for bacterial peptidoglycans. NOD1 detects γ-d-glutamyl-meso-diaminopimelic acid (iE-DAP) from Gram-negative bacteria, whereas NOD2 senses structurally distinct muramyl dipeptide (MDP) moieties from both Gram-positive and -negative bacteria [30,31]. Recognition of respective ligands by NOD1 and NOD2 activates NF-κB and MAPK pathways to drive proinflammatory cytokine and antimicrobial peptide production. Rare gain-of-function variants in NOD2 underlie Blau syndrome and Yao syndrome, both characterized by granulomatous dermatitis and systemic inflammation, underscoring the pathogenic potential of dysregulated NOD2 signaling [32].

NLRP1 and NLRP3, together with absent in melanoma 2 (AIM2), are well-characterized inflammasome sensors in keratinocytes and skin-infiltrating immune cells, serving as intracellular sentinels against microbial infection and cellular stress in skin [33,34]. These proteins detect distinct DAMPs and PAMPs to trigger the assembly of inflammasome complexes, facilitating a diverse recognition of danger signals arising from both external insults and internal damage. NLRP1 inflammasome is activated by ribotoxic stress, reductive and protein folding stress, dsRNA and viral proteases [35,36]. In contrast, NLRP3 is activated by a plethora of stimuli including self-derived and foreign-derived DAMPs, as well as bacterial, viral and fungal PAMPs [37]. It is suggested that NLRP3 senses common cellular stress induced by DAMPs and PAMPs rather than direct binding to them [37,38]. On the other hand, AIM2 mainly detects cytosolic dsDNA released by bacterium or viruses, as well as self-derived dsDNA such as mitochondrial DNA [39]. NLRP10, however, promotes inflammasome activation by recognizing mitochondrial damage independent of mitochondrial DNA [24]. The diverse recognition patterns of inflammasome sensors highlight the complexity of skin innate immunity to provide surveillance against a variety of pathogens and cellular damages.

Pathogenic gain-of-function mutations in NLRP1 are linked to diverse cutaneous and systemic phenotypes, including multiple self-healing palmoplantar carcinoma (MSPC), familial keratosis lichenoides chronica (FKLC), vitiligo, melanoma, and multiple sclerosis [35]. Activating NLRP3 mutations cause cryopyrin-associated periodic syndromes (CAPS), featuring recurrent fevers, urticaria-like rash, and joint diseases [37,40]. Of particular relevance to AD, NLRP10 has emerged as the sole NLR family member with a reproducible genetic association with AD [4,9,10,14,15,16,17,41,42,43]. Beyond its genetic linkage, NLRP10 functions as a critical regulator of keratinocyte survival, differentiation and barrier integrity [4], as well as a scaffold protein for anti-bacterial Nodosome signaling [23] and a mitochondrial damage sensor [24]. Collectively, these findings identify NLRP10 as a pivotal regulator of skin homeostasis and inflammation. Accordingly, this review concentrates on examining its developing significance in skin biology and its relevance to AD.

**Table 1 ijms-26-09623-t001:** Overview of human NLRs in skin biology, homeostasis, and disease genetics.

NLRs	Expression Profile in Skin	Role in Skin	Human Genetic Association with Skin Diseases	References
CIITA	Antigen-presenting cells; Langerhans cells; inducible in keratinocytes (e.g., by IFN-γ)	Drives MHC class II expression	No known variants	[27]
NAIP	Keratinocytes, immune cells	Anti-apoptotic protein by inhibiting caspase activity; sensor for intracellular bacterial factors with NLRC4 (minimal skin expression)	No known variants	[26,44]
NOD1	Immune cells; dermal fibroblasts; inducible in keratinocytes	Intracellular bacterial PGN (iE-DAP) sensor; activates NF-κB and MAPK pathways; host defense against bacteria	* No known variants	[45]
NOD2	Immune cells and basal keratinocytes	Intracellular bacterial PGN (MDP) sensor; activates NF-κB and MAPK pathways; host defense against bacteria	* GOF mutations cause Blau syndrome and Yao syndrome	[32,45]
NLRC5	Immune cells; keratinocytes; dermal fibroblasts.	Transcriptional co-activator for MHC class I genes; may dampen NF-κB and IFN-I responses in some contexts	No known variants	[28]
NLRX1	Ubiquitously expressed	Mitochondrial NLR; regulates ROS and apoptosis; suppresses RIG-I/MAVS, NF-κB and inflammasome	No known variants	[29]
NLRP1	Highly expressed in keratinocytes; immune cells	Major inflammasome sensor in keratinocytes; drives IL-1β/IL-18 release and pyroptosis	* GOF mutations cause MSPC, FKLC, corneal intraepithelial dyskeratosis, vitiligo, NAIAD, multiple keratoacanthomas, multiple sclerosis and melanoma	[35]
NLRP2	Keratinocytes; immune cells	Can form an inflammasome; inhibit NF-κB and interferon signaling	No known variants	[46]
NLRP3	Predominantly in innate immune cells; low in keratinocytes; also expressed in sebocytes	Inflammasome sensor; drives IL-1β/IL-18 release and pyroptosis	* GOF mutations cause CAPS: FCAS1, MWS, NOMID	[37,40]
NLRP10	Keratinocytes; dermal fibroblasts	Regulator of keratinocyte survival, differentiation and barrier function; immune sensor for mitochondrial damage; scaffold for anti-bacterial Nodosome signaling	Multiple variants in NLR10 are associated with AD.	[4,23,24,47]

* Genetic association with AD of variants in the gene locus has been reported in the literature, but the statistics did not reach genome-wide significance. Therefore, those genetic analyses are not included in this review.

## 3. Genetic Association of NLRP10 with Atopic Dermatitis

Genome-wide association studies (GWAS) have established a genetic connection between NLRP10 and susceptibility to AD [9,10,14,15,16,17,41,42,43]. Notably, rs59039403-A, a missense variant encoding the R243W mutation of NLRP10, is significantly associated with protection in AD in east Asian population across multiple independent studies [15,16,17,41,42,43]. NLRP10 R243 is located next to a conserved Walker B motif within the NACHT domain that is necessary for the ATPase activity [24,48]. It is possible that the R243W mutation may influence ATPase activity, affect protein conformation, alter protein–protein interaction, and modulate NLRP10 function. The exact mechanism by which R243W confers protection in AD warrants further investigation. Another intergenic single nucleotide polymorphism (SNP) rs878860-G near the NLRP10 locus is reported as a risk variant in AD [14]. This variant is located within a distal enhancer region that interacts with NLRP10, as identified by targeted chromosome conformation capture studies in differentiating human keratinocytes [49]. Intriguingly, this AD risk variant results in increased similarity to the GATA consensus sequence, which in turn suppresses NLRP10 promoter activity in both growing and differentiated keratinocytes [50]. Additionally, analysis of lesional AD skin biopsy revealed that NLRP10 expression is markedly reduced in AD skin as compared to healthy skin [4,51,52,53,54]. Collectively, these genetic findings and clinical observations link NLRP10 to AD pathogenesis and suggest that downregulation of NLRP10 expression heightens AD risk.

## 4. Molecular Functions of NLRP10

### 4.1. Functional Divergence Between Mouse and Human NLRP10

NLRP10 was initially identified as an Apaf-1-like protein based on homology to NLRP3 and was characterized as an inhibitor of ASC and caspase-1 [55]. *Nlrp10*-deficient mice have been generated and were reported to be defective in mounting an adaptive immune response to various adjuvants due to defects in dendritic cell migration [56]. However, this phenotype was later attributed to a coincidental loss of DOCK8 instead of NLRP10 deficiency [57], suggesting caution is needed when interpreting results obtained from *Nlrp10*-deficient mice. Independent laboratories have generated additional *Nlrp10*^−/−^ mice strains without the concurrent *Dock8* mutation and reported divergent and sometimes conflicting roles of NLRP10 in mediating both proinflammatory and anti-inflammatory function [58,59,60,61,62], reflecting the complexity of its biology and the context-dependent nature of its activity. Mouse models have been widely used to provide valuable insights into human gene functions. However, it should be noted that there are significant differences in the immune system of mice and humans, and findings from murine studies might not always translate to human protein functions [63]. It is not uncommon that NLR family proteins display species-specific regulation and function. For example, human NLRP1 diverges from its murine counterpart in the structural domain composition, spectrum of activating ligands, and distribution across tissues [64,65]. Protein sequence alignment between human and mouse NLRP10 highlights significant divergence of human NLRP10 from its murine homolog in putative functional domains including the PYD and NACHT domain (Figure 2). In addition, structural analysis has indicated that the conformation of human NLRP10 PYD is distinct from that of mouse NLRP10, and the homotypic PYD-PYD interactions of the human NLRP10 differs from its mouse counterpart [66]. This divergence in amino acid sequence and protein structure suggests that the biochemical properties, signaling pathways, or propensity to form protein complexes of the human NLRP10 may be different from the murine homolog. Therefore, extrapolating functional outcomes from mouse experiments to humans requires caution, as distinct molecular interactions could influence cellular responses differently between species. This highlights the necessity for direct investigation of human NLRP10 to fully understand its biological roles and potential relevance in human disease contexts.

### 4.2. NLRP10 as a Negative Regulator of Keratinocyte Cell Death

NLRP10 was initially suggested as a negative regulator of caspase-1-mediated pyroptosis and IL-1β release in reconstituted HEK293T cells overexpressing NLRP10 [47,55]. However, overexpression systems in non-physiological conditions or non-native cell types may not accurately recapitulate the true biological functions of NLRP10 in human skin [67]. To ensure physiological relevance, CRISPR/Cas9-mediated NLRP10 KO primary normal human epidermal keratinocytes (NHEKs) were generated to investigate the function of NLRP10 [4]. NLRP10-deficient primary human keratinocytes displayed increased cell death at baseline condition and were more sensitive to various apoptosis and pyroptosis stimuli [4]. Biochemical analysis demonstrated that NLRP10 interacts with key effector caspases of pyroptosis and apoptosis including caspase-1 and caspase-8. NLRP10 has been shown to inhibit caspase-1 autoprocessing by binding to the catalytic domain of caspase-1 [47]. In contrast to NLRP3, which requires complex formation with the adaptor protein Apoptosis-associated speck-like protein containing a CARD (ASC) as a platform to recruit and activate caspase-1 [68], NLRP10 may directly interact with caspase-1 [4,47,55]. This direct interaction could prevent the oligomerization of caspase-1 and suppress keratinocyte pyroptosis (Figure 3A). On the other hand, NLRP10 binds to caspase-8 pro-domain, the tandem death-effector domains (tDEDs), which mediates its recruitment to the death-inducing signaling complex (DISC) [4]. This interaction sequesters caspase-8, reduces local caspase-8 concentration recruited to the DISC and limits caspase-8 activation and subsequent apoptosis induction (Figure 3A).

Keratinocyte cell death contributes to spongiosis and epidermal barrier disruption in AD, especially in acute and subacute lesions [69]. Skin-infiltrating T cells drive keratinocyte cell death by two main mechanisms: Fas ligand (FasL) engaging Fas receptors on keratinocyte through direct cell–cell contact [70], and secretion of pro-apoptotic inflammatory cytokines such as interferon-γ (IFN-γ), tumor necrosis factor-α (TNF-α), TNF-related apoptosis-inducing ligand (TRAIL) and TNF-like weak inducer of apoptosis (TWEAK) [71,72,73]. Apoptotic keratinocytes lose their cell–cell adhesion due to cleavage of adherens junction protein E-cadherin and desmosomal cadherins, resulting in weakened cohesion among keratinocytes, characteristic spongiotic changes in eczema lesions and compromised skin barrier function [74,75]. Therapeutic strategies that modulate NLRP10 functions to inhibit keratinocyte cell death will ameliorate cell death-induced spongiosis and contribute to the restoration of skin barrier integrity during the acute and subacute phases of AD.

### 4.3. NLRP10 as a Mitochondrial Damage Sensor

NLRP10 can also assemble an active inflammasome with ASC and caspase-1 in the presence of mitochondrial damage, inducing gasdermin D-dependent pyroptosis and release of mature IL-1β and IL-18 in keratinocytes (Figure 3B) [24,58]. Unlike the AIM2 inflammasome sensing mitochondrial DNA (mtDNA), NLRP10 monitors mitochondrial integrity independently of mtDNA, suggesting that it recognizes distinct molecular entities from damaged mitochondria. The exact identity of such DAMP remains unknown. The finding that NLRP10 can still form an active inflammasome in the absence of an LRR domain suggests that LRR domain is not entirely essential for inflammasome assembly. It is unclear whether other accessory proteins are recruited to facilitate the inflammasome formation, in a way similar to the role of NEK7 in NLRP3 inflammasome [76,77]. It is of significant interest to determine both inactive and active oligomerized structures of the NLRP10 inflammasome complex for better understanding of its activation and regulation.

In contrast to previous reports showing that NLRP10 interacts with ASC and inhibits ASC-dependent NF-κB activation and apoptosis under resting condition [47], NLRP10 was reported to colocalize with ASC upon mitochondrial damage in m-3M3FBS- and thapsigargin-treated cells [24]. It remains important to further investigate the cause of such discrepancy and whether NLRP10 inflammasome is activated under physiological and pathological conditions that cause mitochondrial damage in the skin such as UV irradiation-induced mitochondrial damage and environmental stimulants-triggered excessive ROS production [78].

It is reported that AD skin shows increased expression of superoxide dismutase 2 (SOD2), lipid peroxidation, and cytochrome c in the epidermis, indicating elevated oxidative damage and mitochondrial stress [79]. Consistently, quantitative proteomics analysis of lesional and non-lesional AD epidermis revealed impaired activity of the NRF2-antioxidant pathway and reduced expression of mitochondrial proteins involved in key metabolic pathways in both lesional and non-lesional AD compared to healthy donors [80]. A recent report indicated that mitochondrial activity is already upregulated in non-lesional AD keratinocytes, contributing to high p65 NF-κB activity, abnormal lamellar bodies and cellular damage [81]. These clinical data underscore an important impact of mitochondrial dysfunction and damage on AD pathogenesis, suggesting restoring mitochondrial function as a potential strategy for AD treatment [82]. The identification of NLRP10 as a mitochondrial damage sensor raises compelling questions about whether NLRP10 inflammasome plays any role in AD and how mitochondrial stress pathways in keratinocytes shape inflammation in AD.

### 4.4. NLRP10 Facilitates Epidermal Barrier Function

One of the most important functions of the epidermis is to form the epidermal barrier to prevent water loss and protect against environmental insults. Loss of the skin barrier is often an important feature of inflammatory skin diseases including AD [6]. Epidermal barrier, primarily formed by the stratum corneum, requires proper keratinocyte differentiation and lipid matrix integrity [7]. NLRP10 has been identified as a critical regulator of these processes as shown in a three-dimensional human skin equivalent (HSE) organoid culture system [4], which closely resembles physiological features of human skin.

In HSEs, NLRP10 is required for the full development of basal and suprabasal epidermal layers [4]. NLRP10 also promotes the formation of keratinocyte cell–cell junction and expression of key epidermal terminal differentiation proteins including filaggrin and caspase-14. Untargeted lipidomics analysis revealed that NLRP10 is essential for the production of ceramide species, which are critical components of the barrier function. NLRP10-deficient HSEs also display increased biotin paracellular permeability, suggesting barrier dysfunction in the absence of NLRP10 [4]. Together, these characterize an important role of NLRP10 in promoting epidermal differentiation and maintaining epidermal barrier function. Mechanistically, NLRP10 supports epidermal differentiation and barrier function by stabilizing p63, a master transcription factor essential for keratinocyte differentiation (Figure 3C) [83,84,85]. NLRP10 physically interacts with p63 and enhances its protein stability and half-life [4]. This finding highlights the versatility of NLRP10 as a signaling hub to mediate important cellular functions. It is not unprecedented that a cytosolic protein could interact with and stabilize a transcription factor whose role is mainly in the nucleus. A well-known example is that cytosolic 14-3-3σ interacts with p53 and stabilizes p53 expression by antagonizing MDM2-mediated p53 ubiquitination and degradation [86]. NLRP10 might interfere with the activities of p63-targeting E3 ligases to enhance p63 stability. It remains to be determined which E3 ligase is responsible for p63 degradation in keratinocytes and how NLRP10 alters its function in mediating p63 ubiquitination and degradation.

P63, a master transcription factor for keratinocyte proliferation and differentiation, is implicated in AD pathogenesis. In the epidermis, p63 is highly expressed in the basal keratinocytes, where it maintains the stemness of progenitor and stem cells but also initiates terminal differentiation program [83,84]. During epidermal differentiation, p63 regulates the expression of a plethora of target genes through temporal- and spatial-specific active enhancers [87], including important epidermal differentiation regulators IKKα [88,89] and ZNF750 [90]. NLRP10 selectively regulates the p63-ZNF750 axis while sparing the IKKα pathway [4], suggesting that NLRP10 might also shape p63-dependent transcriptome to promote epidermal differentiation and barrier function, in addition to enhancing p63 stability. Interestingly, transcription factor ZNF750 promotes terminal epidermal differentiation through the induction of KLF4 [90] and also mitigates skin inflammation by down-regulating the expression of various pattern recognition receptors [91], suggesting the NLRP10-p63-ZNF750 axis might be involved in both keratinocyte differentiation and inflammation regulation. On the other hand, p63 is dysregulated in AD epidermis [92]. Prolonged exposure to type 2 cytokines, notably IL-4 and IL-13, which are highly elevated in AD, can increase p63 levels and block proper keratinocyte differentiation [93,94]. Paradoxically, dysregulated p63 contributes to the proinflammatory program through the activation of NF-kB pathway [95,96,97]. Despite the reported upregulation of p63 protein in AD, the p63 target gene ZNF750 appears to be downregulated in AD [98]. It remains underexplored which p63 direct target genes mediate such pathogenic changes in AD. A targeted enhancement of p63 signaling that preserves its role in epidermal differentiation and barrier formation, while avoiding its proinflammatory effects, may represent an effective therapeutic strategy for restoring skin barrier integrity. The NLRP10–p63–ZNF750 axis may serve as a valuable target for such intervention.

### 4.5. NLRP10 as a Regulator of the Nodosome to Control Bacterial Infection

NLRP10 is also highly expressed in dermal fibroblasts besides epidermal keratinocytes in the human skin [4,23]. Studies have demonstrated a role of NLRP10 in augmenting proinflammatory responses against bacterial infection by regulating the NOD1-Nodosome and NF-κB pathways in primary human fibroblasts [23,67,99]. NOD1 is a well-characterized NLR that detects iE-DAP-containing peptidoglycan from many Gram-negative bacteria and certain Gram-positive bacteria strains [30]. Ligand recognition steers the auto-inhibited NOD1 monomer into an open conformation, leading to NOD1 oligomerization via its NACHT domain and recruitment of the RIP2 kinase through homotypic CARD-CARD interactions, which initiates a signaling cascade that activates NF-κB and MAPK pathways [30,31]. Using an invasive Gram-negative *Shigella flexneri* infection model and siRNA-based NLRP10 knockdown, Lautz et al. showed that NLRP10 and NOD1 are recruited to the bacterial entry site in primary dermal fibroblasts [23]. This recruitment is dependent on functional ATPase activity and the PYD of NLRP10, which contribute to high affinity binding with NOD1 and downstream signaling molecules such as RIP2, TAK1, and NEMO. NLRP10 possibly functions as a scaffold protein to promote the assembly and stability of the active NOD1-Nodosome complex, leading to enhanced activation of NF-κB and p38 MAPK and the production of proinflammatory cytokines including IL-6 and IL-8, critical for controlling invading bacteria (Figure 3D) [23]. In addition, NLRP10 was also found to enhance NF-κB activation by destabilizing Abin-1, a negative regulator of NF-κB, during *Shigella flexneri* infection (Figure 3D) [99]. Abin-1 inhibits NF-κB by linking ubiquitin-editing enzyme A20 to an active, polyubiquitinated NF-κB mediator protein such as NEMO, modifying the ubiquitination status of NEMO and inhibiting NF-κB, or by competing with NF-κB mediators for binding to polyubiquitinated signaling proteins and preventing their activation [100]. NLRP10 forms a complex with Abin-1 via its NACHT domain, leading to Abin-1 ubiquitination and degradation in a dose-dependent manner [99]. The reduction in Abin-1 levels removes the inhibitory constraint on NF-κB, thereby amplifying proinflammatory signaling and cytokine production for controlling bacterial infection. The involvement of NLRP10 in Nodosome and NF-kB signaling during bacterial infection suggests NLRP10 as an important regulator for controlling skin microbial infection.

A hallmark of AD is skin microbial dysbiosis, with a loss of microbial diversity and overgrowth of *Staphylococcus aureus* (*S. aureus*) on the skin [101]. *S. aureus* dominates in active AD lesions, and its prevalence is positively correlated with disease severity [102]. Recurrent *S. aureus* infection contributes to AD development and disease exacerbation [103]. *S. aureus* expresses several immune-modulating molecules to stimulate keratinocyte proinflammatory responses and activate the immune system [104]. It has also been suggested that *S. aureus* can activate the NLRP1 inflammasome in keratinocytes, leading to the secretion of IL-1β and IL-18 [105], contributing to barrier disruption and type 2 inflammation [106,107]. This correlates with increased expression of NLRP1, IL-18, ASC and caspase-1, as well as a higher degree of caspase-1 activation in AD skin samples [105]. Toxins and proteases produced by *S. aureus* also cause damage to keratinocytes and lead to epidermal barrier dysfunction [103,104]. *S. aureus* can be recognized by host PRRs including the cytosolic NOD proteins to initiate immune responses for host defense and bacterial clearance [108]. In human primary keratinocytes, *S. aureus* infection activates NOD2-driven NF-kB pathway, leading to the production of IL-17C, an epithelial cytokine that induces antimicrobial peptides and inflammatory signals to restrict bacterial growth [109]. As outlined in Table 1, NOD2 senses MDP derived from bacteria, differing from NOD1 ligand iE-DAP [110]. Upon activation, both NOD1 and NOD2 drives the activation of NF-kB and MAPK pathways similarly through RIP2-TAK1-IKK axis [30]. Given that NLRP10 facilitates NOD1 signaling by promoting interaction of NOD1, RIP2, TAK1 and NEMO (Figure 3D), it is plausible that NLRP10 might play a similar scaffolding role for NOD2 signaling in keratinocytes. It remains intriguing to investigate if NLRP10 is involved in the regulation of *S. aureus* activated NOD2 Nodosome and NF-kB signaling, raising the possibility that targeting NLRP10 would serve as a potential approach to promote its ability to enhance anti-bacterial response for the clearance of pathogenic *S. aureus.*

## 5. Therapeutic Strategies Targeting NLRP10

Given the important roles of NLRP10 in maintaining keratinocyte survival and skin barrier function, it represents a promising therapeutic target for AD and other inflammatory skin disorders, especially for patients with down-regulation of NLRP10 expression in the skin. Importantly, human genetic association studies link NLRP10 to AD susceptibility, in alignment with the notion that targets supported by human genetic evidence are more likely to show clinical translatability [111]. An AD risk variant is associated with decreased expression of NLRP10 [50], and AD skins show decreased NLRP10 expression [4,51,52,53,54]. In line with this, NLRP10-deficient HSEs display keratinocyte differentiation defects and barrier dysfunction [4], resembling critical features of AD pathology. Together, these provide a strong rationale for boosting NLRP10 expression in AD. On the other hand, an AD protective variant is related to missense R243W mutation in NLRP10 and possible functional alteration [17], indicating modulating NLRP10 function may also have therapeutic potential. Strategies to modulate NLRP10 expression or activity could enhance keratinocyte survival and restore barrier function in AD, where cell death and disrupted skin barrier contribute to the pathogenesis. Due to the functional discrepancies between mouse and human NLRP10, in vivo validation of this hypothesis will require studies in higher order species such as non-human primates and ultimately humans in clinical trials.

Topical gene delivery of NLRP10 has the potential to restore epidermal NLRP10 expression in atopic dermatitis, where its expression is known to be reduced. The field has witnessed huge progress in topical gene therapy. For example, VYJUVEK is a modified herpes simplex virus type 1 (HSV-1) vector-based gene therapy to deliver a functional copy of the *COL7A1* gene in topical gel formulation, currently approved by the FDA for the treatment of rare genetic disorder dystrophic epidermolysis bullosa (DEB) [112]. It is feasible to deliver NLRP10 gene to AD lesions in a formulation similar to VYJUVEK with high skin tropism, minimal immune activation and no host DNA integration [113]. Alternative non-viral gene delivery methods, such as lipid nanoparticles-based topically formulated “gene creams” [114], hyperbranched polymer-mediated plasmid delivery [115], and other physical gene transfer enhancement techniques [116,117], offer promise of topical gene therapy to restore NLRP10 expression in lesional AD skin. It should be noted that achieving consistent transgene expression in lesional sites might be challenging due to epidermal hyperplasia and skin thickening in AD, even though AD skin typically displays disrupted barrier function [118]. Another key concern is gene dose. Transgene expression level will need to be carefully monitored to ensure over-expressed NLRP10 does not cause unintended responses associated with non-physiological levels of NLRP10. For example, uncontrolled NLRP10 overexpression can cause heightened sensitivity to the activation of NLRP10 inflammasome, leading to over-production of IL-1β and IL-18 and premature cell death via GSDMD-mediated pyroptosis. This may exacerbate inflammation by initiating proinflammatory cytokine signaling cascade and inducing epidermal hyperplasia, reminiscent of skin inflammatory syndromes caused by NLRP1 gain-of-function mutations [119].

Small molecules targeting NLRP10 represents another attractive opportunity to modulate NLRP10 activity to enhance keratinocyte viability and skin barrier function. The molecular function of NLRP10 relies on its role as a scaffold and a signaling hub to modulate signal transduction of other important signaling pathways such as caspase-8-dependent apoptosis and p63-dependent keratinocyte differentiation [4]. Protein conformational change plays an important regulatory role in controlling NLRP protein–protein interaction interface and subsequent scaffolding function [120]. It will be helpful to understand the protein structure of the AD protective NLRP10 missense variant R243W and its interactome. A small molecule that stabilizes such conformation will likely resemble the biochemical properties of the R243W variant and confer protection against AD. A potential challenge of this approach is to identify druggable pocket within NLRP10 suitable for allosteric modulation to stabilize or disrupt protein–protein interaction. Although full-length NLRP10 structure is not yet available, compounds that bind to an allosteric pocket in the NACHT domain of NLRP3 have been identified [121], demonstrating the feasibility of developing allosteric small molecule modulators of NLRP family proteins. DNA-encoded chemical library (DEL) technology is a powerful platform to allow efficient screening of billions of small molecule compounds to identify specific binders to a target protein [122,123]. A possible hit-finding strategy for NLRP10 would be to perform DEL screens to identify NLRP10 binders, followed by high throughput functional screens in epidermal barrier assays such as the 96-well Electric Cell–substrate Impedance Sensing Transepithelial electrical resistance system (ECIS TEER96) for two-dimensional cellular assay [124] and three-dimensional human skin equivalent assay [4]. The proinflammatory side of NLRP10 function in mitochondrial damage-induced inflammasome and bacterial infection-activated Nodosome necessitates precise counter-screening of compounds to avoid unintended excessive inflammation.

Targeting NLRP10 represents a mechanistically distinct strategy compared to existing AD treatments. Current standard-of-care therapies include topical corticosteroids and calcineurin inhibitors, biologics such as dupilumab, tralokinumab, nemolizumab and lebrikizumab, and small-molecule PDE4 and JAK inhibitors [125]. These treatments primarily suppress inflammatory pathways, dampening hyperactive immune response in diseased skin. In contrast, NLRP10-based interventions aim to enhance keratinocyte survival and restore epidermal barrier integrity, addressing an intrinsic pathogenic mechanism in epidermal keratinocytes. NLRP10-targeted therapies would complement existing immunomodulatory treatments by targeting the barrier defect at the core of AD pathogenesis, potentially offering more durable disease modification and better clinical benefits in combination with current AD management strategies.

## 6. Conclusions

NLRP10 is a versatile regulator of skin homeostasis, coordinating key biological processes such as epidermal keratinocyte survival, differentiation, and barrier function. It also contributes to antimicrobial immune responses in dermal fibroblasts. Its ability to inhibit caspase-8-mediated apoptosis and stabilize p63 underscores its role in promoting keratinocyte survival, differentiation and barrier function, while its inflammasome-forming capacity and contribution to Nodosome signaling link it to innate immune responses in the skin. NLRP10 is genetically associated with AD, highlighting its potential as a therapeutic target. Continued research into the molecular regulation and context-specific functions of NLRP10 will pave the way for novel treatments for inflammatory skin disorders including AD and beyond.

## Figures and Tables

**Figure 1 ijms-26-09623-f001:**
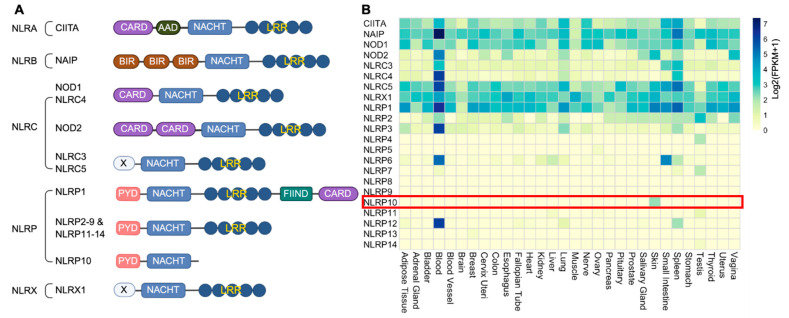
Domain structure and expression pattern of NLRs. (**A**) Human NLRs are categorized into five subfamilies based on the N-terminal effector domains: NLRA, NLRB, NLRC, NLRP and NLRX. CARD, caspase recruitment domain; AAD, acidic activation domain; NACHT, nucleotide-binding and oligomerization domain; LRR, leucine-rich repeat; BIR, baculoviral inhibition of apoptosis protein repeat domain; FIIND, domain with function to find; PYD, pyrin domain; X, unknown domain. NLRP10 is the only member without an LRR domain. (**B**) Heatmap of expression of each NLR across human tissues. Data are acquired from GTEX and represented as Log2(FPKM + 1). Highlighted is skin-restricted expression of NLRP10.

**Figure 2 ijms-26-09623-f002:**
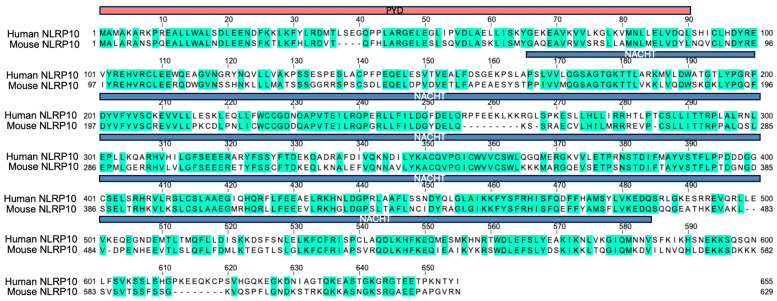
Sequence alignment of human and mouse NLRP10. Protein sequence alignment of human and mouse NLRP10. Identical amino acids are highlighted in green, and dash lines indicate gaps. The PYD domain is shown in red and the NACHT domain is shown in blue. Overall sequence identity between human and mouse NLRP10 is around 60%.

**Figure 3 ijms-26-09623-f003:**
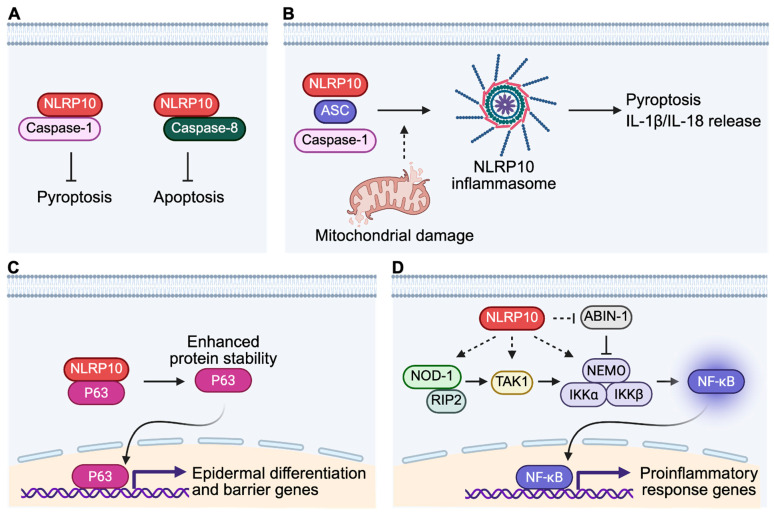
Molecular functions of NLRP10 in human skin. (**A**) NLRP10 binds to caspase-1 and caspase-8 and inhibits caspase-mediated pyroptosis and apoptosis, respectively. (**B**) NLRP10 forms an active inflammasome together with ASC and caspase-1 upon sensing mitochondrial damage. Activation of NLRP10 inflammasome leads to pyroptosis and release of mature IL-1β and IL-18. (**C**) NLRP10 interacts with p63, leading to enhanced protein stability of p63, which is a master regulator of epidermal differentiation and formation of the epidermal barrier. (**D**) NLRP10 functions as a scaffold protein to promote NOD1 Nodosome signaling in dermal fibroblasts. NLRP10 also promotes degradation of ABIN-1, a suppressor of the NF-κB signaling, to facilitate anti-bacterial proinflammatory response.

## Data Availability

No new data were created. NLR expression data were downloaded from GTEX with Qiagen OmicSoft Studio software (v12.8). The raw data are provided in the Appendix A. Heatmap was created in R (version 4.4.1).

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
