# Peer review of "Targeting NLRP10 in Atopic Dermatitis: An Emerging Strategy to Modulate Epidermal Cell Death and Barrier Function"

_ijms, 2025, doi:10.3390/ijms26199623_

Round 1

Reviewer 1 Report

Comments and Suggestions for Authors

Brief summary:

This is a solid, focused mini-review that effectively synthesizes emerging data on NLRP10 in AD, focusing on physiology and pathology of NLRP10 in AD. It is strongest as a mechanistic perspective piece. However, its relevance as a comprehensive review is limited by narrow scope, missing broader AD genetics/therapeutics, and speculative translational discussion. This is of interest to individuals interested in NLR biology and novel AD targets. However, it needs improvement in comparative context, integration with broader AD genetics, and balanced discussion of therapeutic feasibility.

General concept comments:

Completeness: There are some missing elements that should be incorporated into the article. This includes in the following areas;

Genetic evidence: https://doi.org/10.1038/s41467-023-41180-2 and related: https://doi.org/10.1097/ACI.0000000000001005

Please compare NRLP10 to other inflammasomes related to skin and AD

Please discuss in briefly drugability challenges and delivery hurdles of for gene therapy/small molecules. And compare to other AD treatments for reference.

Please discuss more info related to skin microbiome dysbiosis in AD and how this relates to NRLP10

Please include more information about how NLRP10 inflammasome could exacerbate inflammation, and tie this into how causational evidence of upregulation NRLP10 is required.

another suggestion to increase completeness: a comparative table between NLRs in skin

Relevance to topic:

Highly relevant article due to;

Explains structure and uniqueness

Summarizes genetic associations with AD

Discusses potential mechanistic roles

Connects AD pathology

Discusses therapeutic strategies

Gap in knowledge:

Comparative context with other NLRs and comparative role in AD

Including other genetic studies

Contextualizing NLRP10s relative importance

Further discussion of functional contradictions. This should be further analyzed and discussed.

Therapeutic translation is speculative. Lack of in vivo data and potential risks should be discussed. And compared/contrasted to current methods.

Discussion of skin microbiome dysbiosis

Appropriateness of references: references are appropriate. many are recent and primary sources, including contradictory reports. There does not appear to be any evidence of excessive self-citations. A few additions in the above areas could be added.

Ethics: acceptable.

Conflict of Interest: acceptable, with no apparent reason for concern.

Data availability: data is provided and explained

Evaluation:

Novelty: High

Scope:  Low

Significance: High

Quality: Medium

Scientific Soundness: High

Interest to Readers: High

Overall Merit: Medium

English Level: Overall, the quality of writing is very high. However, the abstract needs to be tightened up. As this is the most important part to get viewers attention and to encourage individuals to read the article, it is a shame the abstract has some areas that need to be improved for flow or need to be worded in a more scientific manner. For example: Compared to other NLRs, the function of NLRP10 is less understood especially its physiological role in the skin.

Specific Comments:

Figure 3 should be contained within section 3, not section 2.

Alternatively, figure 3 could be split into each subfigure, and expanded on for each section that each subfigure is mentioned. This would likely help.

Overall Recommendation:

This is high quality mini-review that needs to be expanded a bit first.

My recommendation, reconsider after major revisions

Reviewer 2 Report

Comments and Suggestions for Authors

This manuscript presents a comprehensive and insightful overview of the multifaceted roles of NLRP10 in skin biology, with a particular emphasis on its relevance to atopic dermatitis (AD) and other inflammatory skin disorders. The authors offer a detailed examination of the molecular mechanisms through which NLRP10 contributes to keratinocyte survival and skin barrier integrity, positioning it as a promising therapeutic target.The discussion is well-structured and grounded in both mechanistic and translational perspectives. The manuscript underscores the dual opportunities of therapeutic intervention, either by enhancing NLRP10 expression or modulating its functional conformation.Drawing parallels with FDA-approved therapies such as VYJUVEK adds credibility and translational weight to the proposed strategies. The authors also demonstrate awareness of safety considerations, emphasizing the need for controlled transgene expression to avoid off-target effects as well as proinflammatory risks associated with NLRP10 modulation.

Overall, this work offers a thorough analysis of NLRP10 as a therapeutic target, bridging molecular insights with translational applications. It will be of significant interest to researchers in dermatology, immunology, and drug development, and is acceptable for publication.

Based on the provided overview of NLRP10 mechanisms in keratinocytes, could you formulate a final recommendation at the end of the Discussion section regarding the design of a screening strategy to identify candidate compounds targeting NLRP10?

Round 2

Reviewer 1 Report

Comments and Suggestions for Authors

End of the abstract, first person is used. Do not use first person. Please rewrite… for example: “Furthermore, therapeutic opportunities … are highlighted.”

Section 2, the table has been added related to other NLPR10, but the table is large and chunky now, with little discussion in the written. The table should contain less writing, but still the same information, just more succinctly. Furthermore, the section does not adequately compare NLRP10 to others. I feel there should be at least one additional paragraph that discusses the main differences of NLRP10 with the other NLRPs (focusing on ones that are most related and relevant).  I feel section 2 needs to be expanded. While others NLPR10s are briefly mentioned later on, outside of this section, I think briefly comparing NLPR10 to other NLRPs in a paragraph or two would be helpful, instead of just telling readers to look at the clunky table.

What I am saying is more of a stylistic change, as much information is given in the data.

As such, I’m suggesting that IJMS accept upon minor revisions.

Good work Author.
